# Tiny Attention: A simple yet effective method for learning contextual word embeddings

**Renjith P Ravindran, Kavi Narayana Murthy**
School of Computer and Information Sciences,
University of Hyderabad, Hyderabad, India
*renjithravindran88@gmail.com, knmuh@yahoo.com*

## Abstract

Contextual Word Embedding (CWE) obtained via the Attention Mechanism in Transformer (AMT) models is one of the key drivers of the current revolution in Natural Language Processing. Previous techniques for learning CWEs are not only inferior to AMT but also are largely subpar to the simple bag-of-words baseline. Though there have been many variants of the Transformer model, the attention mechanism itself remains unchanged and is largely opaque. We introduce a new method for leaning CWEs that uses a simple and transparent attention mechanism. Our method is derived from the SVD based Syntagmatic Word Embeddings, which capture word associations. We test our model on the Word-in-Context dataset, and show that it outperforms the simple but tough-to-beat baseline by a substantial margin.

## 1 Introduction

Contextual Word Embeddings (CWE) (Peters et al., 2018) can provide different representations of the same word (type) occurring in different contexts. CWEs capture polysemy in natural languages, unlike vanilla embeddings such as Word2Vec(Mikolov et al., 2013), which provides a single representation and therefore is non-contextual. Non-contextual embeddings can be promoted to a contextual embedding by simply adding together embeddings of all words in a sentence. This sentence bag-of-words technique though simple is considered a tough-to-beat baseline(Arora et al., 2017; Pilehvar & Camacho-Collados, 2018). The CWE that can beat the baseline by a significant margin is the embeddings obtained from Transformer models (Devlin et al., 2018). However, these models are complex, require large scale compute and are difficult to interpret. Here we show that the attention mechanism can be simplified to its essence, ie. by directly computing attention weights.

## 2 Syntagmatic Associations as Attention

Consider the following sentences having the same polysemous word – "They went to the river **bank**" and "The **bank** was short on cash". The meaning of 'bank' is different in the two sentences because in one it is associated with 'river' and in other it is associated with 'cash'. The sentence bag-of-words technique may be improved by selecting only such associated words to be added to the target word. We use Syntagmatic Word Embeddings (SE) (P. Ravindran et al., 2021) to obtain words associated with a target word in a sentential context. Because they capture word associations they are in contrast to paradigmatic embeddings which capture similarity, such as Word2vec. To give an example, 'cat' and 'dog' are paradigmatically related, 'dog' and 'bark' are syntagmatically related. Syntagmatic embedding have been previously used for the task of selectional preference, where the model has to decide the most likely associations given a target word (eg. given 'eat', which is more likely 'chalk' or 'cake'?). SE employs SVD and excels at this task in a purely unsupervised setup. Syntagmatic Embeddings provide two representations per word, one of its left context and other its right context. The degree of association of word B to the right of word A is given by the cosine between the right embedding of word A and the left embedding of word B. Similarly, the cosine between the right embedding of word B and the left embedding of word A, gives the association of A to right of B. Figure 1 gives an example of such associations obtained. These are our so called attention weights.

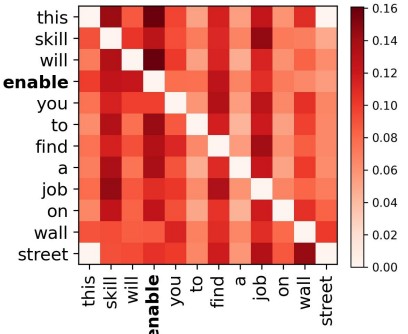

| MODEL | SCORE |
|---|---|
| bow-baseline | 54.7 |
| **syn-sum-3** | **59.0** |
| syn-cat-3 | 52.7 |
| syn-cat-10 | 55.3 |
| par-sum-3 | 55.5 |
| *bow-baseline* | 58.7 |
| *context2vec* | 59.3 |
| *bert-large* | 65.5 |
| *elmo* | 57.7 |
| *human-baseline* | 80.0 |

**Figure 1:** Normalized syntagmatic associations between all pairs of words (except reflexive) in a sentence from WiC. Bold-face marks the target-word.

**Table 1:** WiC test-set scores (accuracy). Models with asterisks have scores taken from the WiC paper.

## 3   OUR MODEL

Given a target word in a sentence, we use the syntagmatic embeddings of (P. Ravindran et al., 2021) to obtain association weights for each of its context words. The top $k$ associated words from both left and right contexts combined are selected. Contextual embedding of the target word is obtained by combining the paradigmatic embeddings of the target word and the $k$ selected context words. The combination function could be either summation or concatenation. Models such as Word2Vec could be used as a source of paradigmatic embeddings.

## 4   EVALUATION

We evaluate our model on the Word-in-Context (WiC) dataset (Pilehvar & Camacho-Collados, 2018). WiC provides a classification task, in which CWE models are required to tell if a target word in two sentences are of the same or different sense. We use the classification method suggested in the paper. The English Wikipedia text dump (2018), is used to train both the syntagmatic and the paradigmatic embeddings. The vocabulary is limited to words occurring at least 500 times. We use the Word2Vec model for the paradigmatic embeddings. The syntagmatic embedding is not a sub-word based model, therefore we can not apply our model on target words that are out of vocabulary. For such cases we simply fallback to bag-of-words of the remaining in-vocabulary words in the sentence. **Models evaluated:** bag-of-word (bow-baseline), our model (syn) with both summation (sum) and concatenation (cat), with a few different values of parameter $k$. To highlight the impact of syntagmatic selection, we include one model with paradigmatic (par) selection.

**Results(Table 1):** Our best model improves over our baseline by more than 4 points. We find that summation is much better than concatenation and that the number of context words to select is 3 ($k$). When selected context words are concatenated, they may need to be aligned based on some criteria across the two sentences. Our model does not provide such alignments. This could explain the poor performance of concatenation. The only non Transformer model better than the baseline reported in WiC paper, is Context2Vec. However it improves by only half a point. We note that the our baseline is considerably lower than the baseline reported in the WiC paper. This is most likely due to the difference in the corpus and the size of the vocabulary (see appendix A).

## 5   CONCLUSION

We have shown that a CWE model with a simple attention mechanism can beat the bag-of-words baseline by a substantial margin. Though we haven't made a direct comparison with a transformer model, we believe it could be competitive as *bert-large* improves over the baseline by about 6 points and ours improve by about 4 points. Bert has multiple layers and multiple attention heads per layer therefore commands a much bigger machinery, ours on the other hand is the humble SVD (Kalman, 1996) and sorting! Future work would look at ways to improve our model (multi-attention) and to fix the limitations of the current study.

URM STATEMENT

The authors acknowledge that at least one key author of this work meets the URM criteria of ICLR 2023 Tiny Papers Track.

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

## A APPENDIX

Following are more details regarding our model.
Corpus: English Wikipedia 2018 dump, uncased, sentence segmented
Vocabulary: about 86k types (mincount = 500)
Syntagmatic Embeddings: window-size = 10, dimensions = 300, term_weight = log
Paradigmatic Embeddings (Word2Vec): skip-gram, window-size = 10, dimensions = 300
Classifier: Simple threshold from grid search (step = 0.02) on WiC dev-set, train-set is not used

