# OpenReview forum: "Tiny Attention: A Simple yet Effective Method for Learning Contextual Word Embeddings"
_ICLR.cc/2023/TinyPapers — Submitted to Tiny Papers @ ICLR 2023_

### Official Review · Reviewer_4NKz · 2023-04-01

**Confidence:** 5

**Summary Of Contributions:**

The paper presents a method to create contextual word embeddings using syntagmatic embedding based attention. The CWEs are evaluated on the WiC dataset and shown to perform better than the BOW baseline.

**Rating:**

Clear, Correct, and Reproducible (CCR): a submission which meets the reviewing criteria

**Strengths And Weaknesses:**

### Strengths ###
- Clarity: The authors have clearly described their methodology of embedding creating using syntagmatic embedding association-based attention and shown preliminary results on the WiC dataset with improvements over the BOW baseline
- Follows Basic Requirements: The paper meets formatting requirements and the page limit.

### Weaknesses ###
- Correctness: Although the authors claim that their method is 'effective' for learning contextual embeddings, they do not evaluate their model on additional WSD datasets to find whether their method can capture a coarser difference in word meaning, although this is a minor weakness for a tiny paper.



**Suggested Changes:**

none

---

### Official Review · Reviewer_6VJd · 2023-04-02

**Confidence:** 4

**Summary Of Contributions:**

The paper introduces a new method for learning contextual word embeddings (CWE) that uses a simple and transparent attention mechanism derived from the SVD-based Syntagmatic Word Embeddings. The authors test their model on the Word-in-Context dataset and show that it outperforms the simple but tough-to-beat bag-of-words baseline by a substantial margin.

**Rating:**

High Potential (HP): a submission which meets the reviewing criteria and has potential to make an impact on the field

**Strengths And Weaknesses:**

Strengths:

The paper proposes a new method for learning contextual word embeddings that uses a simple and transparent attention mechanism derived from syntagmatic word embeddings.
The paper provides a clear and concise explanation of the proposed method.
Their approach to the use of SVD-based Syntagmatic Word Embeddings for obtaining word associations shows promise for future research.

Weaknesses:

The paper does not compare the proposed method with state-of-the-art contextual word embedding models like BERT, RoBERTa, and GPT-3.
The paper does not provide any insight into the computational complexity of the proposed method or how it scales with the size of the input corpus.
The paper lacks thorough experimentation with different hyperparameters and evaluation of their impact on the performance of the proposed method.

**Suggested Changes:**

Further experiments and results are needed to solidify their approach.

---

### Meta-Review · Area_Chair_VfiP · 2023-04-06

**Recommendation:** Invite to present (notable)
**Confidence:** 4

**Metareview:**

A new method for learning contextual word embeddings (CWE) using Syntagmatic Word Embeddings based attention is introduced. The CWEs are evaluated on the WiC dataset and shown to perform better than the BOW baseline.

Strengths: New approach, clear and concise explanation, and preliminary results show improvement over baseline.

As a “Tiny” paper, this work fits well. Looking forward to seeing the next stage of this research, especially when compared to transformers etc.



**Summary:**

A new method for learning contextual word embeddings (CWE) using Syntagmatic Word Embeddings based attention is introduced. The CWEs are evaluated on the WiC dataset and shown to perform better than the BOW baseline.

**Comments And Feedback To The Authors:**

See if it is possible to address some of the reviewers' comments without the need for extensive additional work.

**Reason For Not Giving A Higher Recommendation:**

N/A

**Reason For Not Giving A Lower Recommendation:**

Great paper.

---

> ### Author Response · Authors · 2023-05-30
> **Thanks**
>
> We thank the meta reviewers and reviewers for their kind and encouraging feedback.
> As this is a Tiny Paper, we are deferring the comparison of our technique with sota models to a separate full paper.

---

### Decision · Program_Chairs · 2023-04-07

Invite to present (notable)